# Automatically Labeled Data Generation for Large Scale Event Extraction

## Abstract

Modern models of event extraction for tasks like ACE are based on supervised learning of events from small hand-labeled data. However, hand-labeled training data is expensive to produce, in low coverage of event types, and limited in size, which makes supervised methods hard to extract large scale of events for knowledge base population. To solve the data labeling problem, we propose to automatically label training data for event extraction via a world knowledge and linguistic knowledge, which can detect key arguments and trigger words for each event type and employ them to label events in texts automatically. The experimental results show that the quality of our large scale automatically labeled data is competitive with elaborately human-labeled data. And our automatically labeled data can incorporate with human-labeled data, then improve the performance of models learned from these data.

## 1 Introduction

Event Extraction (EE), a challenging task in Information Extraction, aims at detecting and typing events (Event Detection), and extracting arguments with different roles (Argument Identification) from natural-language texts. For example, in the sentence shown in Figure 1, an EE system is expected to identify an *Attack* event triggered by *threw* and extract the corresponding five augments with different roles: *Yesterday* (Role=*Time*), *demonstrators* (Role=*Attacker*), *stones* (Role=*Instrument*), *soldiers* (Role=*Target*), and *Israeli* (Role=*Place*).

To this end, so far most methods ([Nguyen et al.,](#))

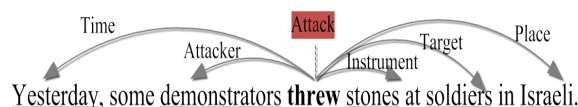

Figure 1: This sentence expresses an *Attack* event triggered by *threw* and containing five arguments.

[2016;](#) [Chen et al., 2015;](#) [Li et al., 2014;](#) [Hong et al.,](#) [2011;](#) [Ji and Grishman, 2008](#)) usually adopted supervised learning paradigm which relies on elaborate human-annotated data, such as ACE 2005[1], to train extractors. Although this paradigm was widely studied, existing approaches still suffer from high costs for manually labeling training data and low coverage of predefined event types. In ACE 2005, all 33 event types are manually predefined and the corresponding event information (including triggers, event types, arguments and their roles) are manually annotated only in 599 English documents since the annotation process is extremely expensive. As Figure 2 shown, nearly 60% of event types in ACE 2005 have less than 100 labeled samples and there are even three event types which have less than ten labeled samples. Moreover, those predefined 33 event types are in low coverage for Natural Language Processing (NLP) applications on large-scale data.

Therefore, for extracting large scale events, especially in open domain scenarios, how to automatically and efficiently generate sufficient training data is an important problem. This paper aims to automatically generate training data for EE, which involves labeling triggers, event types, arguments and their roles. Figure 1 shows an example of labeled sentence. Recent improvements of Distant Supervision (DS) have been proven to be effective to label training data for Relation Extraction (RE), which aims to predict semantic relations between pairs of entities, formulated as $(entity_1, relation, entity_2)$. And DS for RE as-

---
[1]http://projects.ldc.upenn.edu/ace/

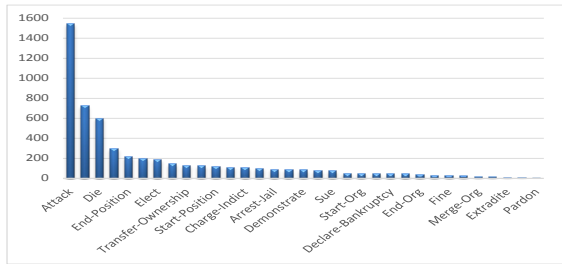

Figure 2: Statistics of ACE 2005 English Data.

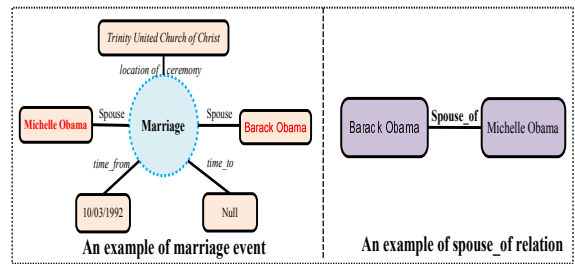

Figure 3: A comparison of events and relations.

sumes that if two entities have a relationship in a known knowledge base, then all sentences that mention these two entities will express that relationship in some way (Mintz et al., 2009). However, when we use DS for RE to EE, we meet following challenges:

**Triggers are not given out in existing knowledge bases**. EE aims to detect an event instance of a specific type and extract their arguments and roles, formulated as ($event\ instance, event\ type$; $role_1, argument_1; role_2, argument_2; ...; role_n, argument_n$), which can be regarded as a kind of multiple or complicated relational data. In Figure 3, the right part shows an example of $spouse\_of$ relation between $Barack\ Obama$ and $Michelle\ Obama$, where two rectangles represent two entities and the edge connecting them represents their relation. DS for RE uses two entities to automatically label training data; In comparison, the left part in Figure 3 shows a *marriage* event of $Barack\ Obama$ and $Michelle\ Obama$, where the dash circle represents the *marriage* event instance of $Barack\ Obama$ and $Michelle\ Obama$, rectangles represent arguments of the event instance, and each edge connecting an argument and the event instance expresses the role of the argument. For example, $Barack\ Obama$ plays a $Spouse$ role in this marriage event instance. It seems that we could use an event instance and an argument to automatically generate training data for argument identification just like DS for RE. However, an event instance is a virtual node in existing knowledge bases and mentioned implicitly in texts. For example, in Freebase, the aforementioned marriage event instance is represented as $m.02nqglv$ (see details in Section 2). Thus we cannot directly use an event instance and an argument, like $m.02nqglv$ and $Barack\ Obama$, to label back in sentences. In ACE event extraction program, an event instance is represented as a trigger word, which is the main word that most clearly represents an event occurrence in sentences, like *threw*

in Figure 1. Following ACE, we can use trigger words to represent event instance, like *married* for *people.marriage* event instance. Unfortunately, triggers are not given out in existing knowledge bases.

To resolve the trigger missing problem mentioned above, we need to discover trigger words before employing distant supervision to automatically label event arguments. Following DS in RE, we could naturally assume that a sentence contains all arguments of an event in the knowledge base tend to express that event, and the verbs occur in these sentences tend to evoke this type of events. However, **arguments for a specific event instance are usually mentioned in multiple sentences**. Simply employing all arguments in the knowledge base to label back in sentences will generate few sentences as training samples. As shown in Table 1, only $0.02\%$ of instances can find all argument mentions in one sentence.

| Event Type | EI# | A# | S# |
|---|---|---|---|
| education.education | 530,538 | 8 | 0 |
| film.film_crew_gig | 252,948 | 3 | 8 |
| people.marriage | 152,276 | 5 | 0 |
| ... | ... | ... | ... |
| military.military_service | 27,933 | 6 | 0 |
| olympics.olympic_medal_honor | 20,790 | 5 | 4 |
| sum of the selected 21 events | 3,870,492 | 100 | 798 |

Table 1: Statistics of events in Freebase. EI# denotes number of event instances in Freebase. A# denotes number of arguments for each event types, and S# indicates number of sentences contain all arguments of each event type in Wikipedia.

To solve above problems, we propose an approach to automatically generate labeled data for large scale EE by jointly using world knowledge (Freebase) and linguistic knowledge (FrameNet). At first, we put forward an approach to prioritize arguments and select key or representative arguments (see details in Section 3.1) for each event type by using Freebase; Secondly, we merely use key arguments to label events and figure out trigger words; Thirdly, an external linguistic knowledge resource, FrameNet, is employed to filter noisy trigger words and expand more triggers; Af-

ter that, we propose a Soft Distant Supervision (SDS) for EE to automatically label training data, which assumes that any sentence containing all key arguments in Freebase and a corresponding trigger word is likely to express that event in some way, and arguments occurring in that sentence are likely to play the corresponding roles in that event. Finally, we evaluate the quality of the automatically labeled training data by both manual and automatic evaluations. In addition, we employ a CNN-based EE approach with multi-instance learning for the automatically labeled data as a baseline for further research on this data. In summary, the contributions of this paper are as follows:

- To our knowledge, it is the first work to automatically label data for large scale EE via world knowledge and linguistic knowledge. All the labeled data in this paper have been released and can be downloaded freely[2].
- We propose an approach to figure out key arguments of an event by using Freebase, and use them to automatically detect events and corresponding trigger words. Moreover, we employ FrameNet to filter noisy triggers and expand more triggers.
- The experimental results show that the quality of our large scale automatically labeled data is competitive with elaborately human-annotated data. Also, our automatically labeled data can augment traditional human-annotated data, which could significantly improve the extraction performance.

## 2 Background

In this paper, we respectively use Freebase as our world knowledge containing event instance and FrameNet as the linguistic knowledge containing trigger information. The articles in Wikipedia are used as unstructured texts to be labeled. To understand our method easily, we first introduce them as follows:

**Freebase** is a semantic knowledge base (Bollacker et al., 2008), which makes use of mediators (also called compound value types, CVTs) to merge multiple values into a single value. As shown in Figure 3, *people.marriage* is one type of CVTs. There are many instances of *people.marriage* and the marriage of *Barack Obama* and *Michelle Obama* is numbered as *m.02nqglv*. *Spouse*, *from*, *to* and *location of ceremony* are

roles of the *people.marriage* CVTs. *Barack Obama*, *Michelle Obama*, *10/3/1992* and *Trinity United Church of Christ* are the values of the instances. In this paper, we regard these CVTs as events, type of CVTs as event type, CVT instances as event instances, values in CVTs as arguments in events and roles of CVTs as the roles of arguments play in the event, respectively. According to the statistics of the Freebase released on $23^{th}$ April, 2015, there are around 1885 CVTs and around 14 million CVTs instances. After filtering out useless and meaningless CVTs, such as CVTs about user profiles and website information, we select 21 types of CVTs with around 3.8 million instances for experiments, which mainly involves events about *education*, *military*, *sports* and so on.

**FrameNet**[3] is a linguistic resource storing information about lexical and predicate argument semantics (Baker et al., 1998). FrameNet contains more than $1,000$ frames and $10,000$ Lexical Units (LUs). Each frame of FrameNet can be taken as a semantic frame of a type of events (Liu et al., 2016). Each frame has a set of lemmas with part of speech tags that can evoke the frame, which are called LUs. For example, *appoint.v* is a LU of *Appointing* frame in FrameNet, which can be mapped to *people.appointment* events in Freebase. And a LUs of the frame plays a similar role as the trigger of an event. Thus we use FrameNet to detect triggers in our automatically data labeling process.

**Wikipedia**[4] that we used was released on January, 2016. All 6.3 million articles in it are used in our experiments. We use Wikipedia because it is relatively up-to-date, and much of the information in Freebase is derived from Wikipedia.

## 3 Method of Generating Training Data

Figure 4 describes the architecture of automatically labeling data, which primarily involves the following four components: (i) Key argument detection, which prioritizes arguments of each event type and selects key arguments for each type of event; (ii) Trigger word detection, which uses key arguments to label sentences that may express events preliminarily, and then detect triggers; (iii) Trigger word filtering and expansion, which uses FrameNet to filter noisy triggers and expand triggers; (iv) Automatically labeled data generation, which uses a SDS to label events in sentences.

---

[2]https://github.com/acl2017submission/event-data

[3]http://framenet.icsi.berkeley.edu
[4]https://www.wikipedia.org/

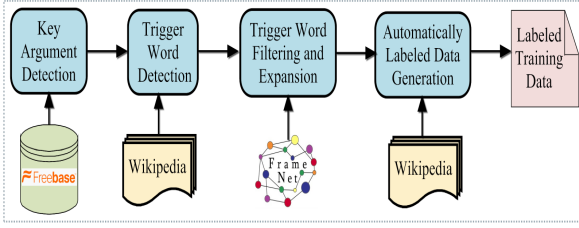

Figure 4: The architecture of automatically labeling training data for large scale event extraction.

### 3.1 Key Argument Detection

This section illustrates how to detect key arguments for each event type via Freebase. Intuitively, arguments of a type of event play different roles. Some arguments play indispensable roles in an event, and serve as vital clues when distinguishing different events. For example, compared with arguments like *time*, *location* and so on, *spouses* are key arguments in a *marriage* event. We call these arguments as key arguments. We propose to use *Key Rate* (KR) to estimate the importance of an argument to a type of event, which is decided by two factors: *Role Saliency* and *Event Relevance*.

**Role Saliency (RS)** reflects the saliency of an argument to represent a specific event instance of a given event type. If we tend to use an argument to distinguish one event instance form other instances of a given event type, this argument will play a salient role in the given event type. We define RS as follows:

$$RS_{ij} = \frac{Count(A_i, ET_j)}{Count(ET_j)} \quad (1)$$

where $RS_{ij}$ is the role saliency of $i$-th argument to $j$-th event type, $Count(A_i, ET_j)$ is the number of $Arguemnt_i$ occurring in all instances of $eventType_j$ in Freebase and $Count(ET_j)$ is the number of instances of $eventType_j$ in Freebase.

**Event Relevance (ER)** reflects the ability in which an argument can be used to discriminate different event types. If an argument occurs in every event type, the argument will have a low event relevance. We propose to compute ER as follows:

$$ER_i = \log \frac{Sum(ET)}{1 + Count(ETCi)} \quad (2)$$

where $ER_i$ is the event relevance of $i$-th argument, $Sum(ET)$ is the number of all event types in knowledge base and $Count(ETC_i)$ is the number of event types containing $i$-th argument. Finally, KR is computed as follows:

$$KR_{ij} = RS_{ij} * ER_i \quad (3)$$

We compute KR for all arguments of each event type, and sort them according to KR. Then we choose top $K$ arguments as key arguments.

### 3.2 Trigger Word Detection

After detecting key arguments for every event types, we use these key arguments to label sentences that may express events in Wikipedia. At first, we use Standford CoreNLP tool[5] to converts the raw Wikipedia texts into a sequence of sentences, attaches NLP annotations (POS tag, NER tag). Finally, we select sentences contains all key arguments of an event instance in Freebase as sentences expressing corresponding events. Then we use these labeled sentences to detect triggers.

In a sentence, a verb tend to express an occurrence of an event. For example, in ACE 2005 English data, there are $60\%$ of events triggered by verbs. As shown in Figure 1, *threw* is a trigger of *Attack* event. Intuitively, if a verb occurs more times than other verbs in the labeled sentences of one event type , the verb tends to trigger this type of event; and if a verb occurs in sentences of every event types, like *is*, the verb will have a low probability to trigger events. Thus we propose *Trigger Candidate Frequency* (TCF) and *Trigger Event Type Frequency* (TETF) to evaluate above two aspects. Finally we employ *Trigger Rate* (TR), which is the product of TCF and TETF to estimate the probability of a verb to be a trigger, which is formulated as follows:

$$TR_{ij} = TCF_{ij} * TETF_i \quad (4)$$

$$TCF_{ij} = \frac{Count(V_i, ETS_j)}{Count(ETS_j)} \quad (5)$$

$$TETF_i = \log \frac{Sum(ET)}{1 + Count(ETI_i)} \quad (6)$$

where $TR_{ij}$ is the trigger rate of $i$-th verb to $j$-th event type, $Count(V_i, ETS_j)$ is the number of sentences, which express $j$-th type of event and contain $i$-th verb, $Count(ETS_j)$ is the number of sentences expressing $j$-th event type, $Count(ETI_i)$ is the number of event types, which have the labeled sentences containing $i$-th verb. Finally, we choose verbs with high $TR$ values as the trigger words for each event type.

### 3.3 Trigger Word Filtering and Expansion

We can obtain an initial verbal trigger lexicon by above trigger word detection. However, this initial trigger lexicon is noisy and merely contains verbal triggers. The nominal triggers like *marriage* are missing. Because the number of nouns in one sentence is usually larger than that of verbs, it is hard to use TR to find nominal triggers. Thus, we propose to use linguistic resource FrameNet to filter

---

[5]http://stanfordnlp.github.io/CoreNLP/

noisy verbal triggers and expand nominal triggers. As the success of word embedding in capturing semantics of words (Turian et al., 2010), we employ word embedding to map the events in Freebase to frames in FrameNet. Specifically, we use the average word embedding of all words in $i$-th Freebase event type name $e_i$ and word embedding of $k$-th lexical units of $j$-th frame $e_{j,k}$ to compute the semantic similarity. Finally, we select the frame contains max similarity of $e_i$ and $e_{j,k}$ as the mapped frame, which can be formulated as follows:

$$frame(i) = \arg\max_j(similarity(e_i, e_{j,k})) \qquad (7)$$

Then, we filter the verb, which is in initial verbal trigger word lexicon and not in the mapping frame. And we use all nouns in the mapped frame to expand trigger lexicon.

### 3.4 Automatically labeled data generation

Finally, we propose a Soft Distant Supervision and use it to automatically generate training data, which assumes that any sentence containing all key arguments in Freebase and a corresponding trigger word is likely to express that event in some way, and arguments occurring in that sentence are likely to play the corresponding roles in that event.

## 4 Method of Event Extraction

In this paper, event extraction is formulated as a two-stage, multi-class classification task. The first stage is called *Event Classification*, which aims to predict whether the key argument candidates participate in a Freebase event. If the key arguments participate a Freebase event, the second stage is conducted, which aims to assign arguments to the event and identify their corresponding roles. We call this stage as *argument classification*. We employ two similar Dynamic Multi-pooling Convolutional Neural Networks with Multi-instance Learning (DMCNNs-MIL) for above two stages. The Dynamic Multi-pooling Convolutional Neural Networks (DMCNNs) is the best reported CNN-based model for event extraction (Chen et al., 2015) by using human-annotated training data. However, our automatically labeled data face a noise problem, which is a intrinsic problem of using DS to construct training data (Hoffmann et al., 2011; Surdeanu et al., 2012). In order to alleviate the wrong label problem, we use Multi-instance Learning (MIL) for two DMCNNs. Because the second stage is more complicated and limited in space, we take the MIL used in arguments classification as an example and describes as follows:

We define all of the parameters for the stage of argument classification to be trained in DMCNNs as $\theta$. Suppose that there are $T$ bags $\{M_1, M_2, ..., M_T\}$ and that the $i$-th bag contains $q_i$ instances (sentences) $M_i = \{m_i^1, m_i^2, ..., m_i^{q_i}\}$, the objective of multi-instance learning is to predict the labels of the unseen bags. In stage of argument classification, we take sentences containing the same argument candidate and triggers with a same event type as a bag and all instances in a bag are considered independently. Given an input instance $m_i^j$, the network with the parameter $\theta$ outputs a vector $O$, where the $r$-th component $O_r$ corresponds to the score associated with argument role $r$. To obtain the conditional probability $p(r|m_i^j, \theta)$, we apply a softmax operation over all argument role types:

$$p(r|m_i^j, \theta) = \frac{e^{o_r}}{\sum_{k=1}^{n} e^{o_k}} \qquad (8)$$

where, $n$ is the number of roles. And the objective of multi-instance learning is to discriminate bags rather than instances. Thus, we define the objective function on the bags. Given all $(T)$ training bags $(M_i, y_i)$, we can define the objective function using cross-entropy at the bag level as follows:

$$J(\theta) = \sum_{i=1}^{T} \log p(y_i|m_i^j, \theta) \qquad (9)$$

where $j$ is constrained as follows:

$$j^* = \arg\max_j p(r|m_i^j, \theta) \quad 1 \le j \le q_i \qquad (10)$$

To compute the network parameter $\theta$, we maximize the log likelihood $J(\theta)$ through stochastic gradient descent over mini-batches with the Adadelta (Zeiler, 2012) update rule.

## 5 Experiments

In this section, we first manually evaluate our automatically labeled data. Then, we conduct automatic evaluations for our labeled data based on ACE corpus and analyze effects of different approaches to automatically label training data. Finally, we shows the performance of DMCNNs-MIL on our automatically labeled data.

### 5.1 Our Automatically Labeled Data

By using the proposed methods, a large set of labeled data could be generated automatically. Table 2 shows the statistics of the five largest automatically labeled events among selected 21 Freebase events. Two hyper parameters, the number of key arguments and the value of TR in our automatically data labeling, are set as 2 and 0.8, by

| Event Type | Freebase Size | Sentences (KA) | Sentences (KA+T) | Examples of argument roles sorted by KR | Examples of triggers |
|---|---|---|---|---|---|
| people.marriage | 152,276 | 56,837 | 26,349 | spouse, spouse, from, to, location | marriage, marry, wed, wedding, couple,..., wife |
| music.group_membership | 239,813 | 90,617 | 20,742 | group, member, start, role, end | musician, singer, sing, sang, sung, concert,..., play |
| education.education | 530,538 | 26,966 | 11,849 | student, institution, degree,..., minor | educate, education, graduate, learn, study,..., student |
| organization.leadership | 43,610 | 5,429 | 3,416 | organization, person, title,..., to | CEO, charge, administer, govern, rule, boss,..., chair |
| olympics.olympic_medal_honor | 20,790 | 4,056 | 2,605 | medalist, olympics, event,..., country | win, winner, tie, victor, gold, silver,..., bronze |
| ... | | | | ... | ... |
| sum of 21 selected events | 3,870,492 | 421,602 | 72,611 | argument1, argument2 ,..., argumentN | trigger1, trigger2, trigger3, ... , triggerN |

Table 2: The statistics of five largest automatically labeled events in selected 21 Freebase events, with their size of instances in Freebase, sentences labeled with key argument (KA) and KA + Triggers(T), examples of arguments roles sorted by KR and examples of triggers.

grid search respectively. When we merely use two key arguments to label data, we will obtain 421, 602 labeled sentences. However, these sentences miss labeling triggers. Thus, we leverage these rough labeled data and FrameNet to find triggers and use SDS to generate labeled data. Finally, 72, 611 labeled sentences are generated automatically. Compared with nearly 6, 000 human annotated labeled sentence in ACE, our method can automatically generate large scale labeled training data.

## 5.2 Manual Evaluations of Labeled Data

##001  He is the uncle of [**Amal Clooney**], [**wife**] of the actor [**George Clooney**].
**Trigger:** wife    **Event Type:** Marriage       **MannalAnotate[Y/N]:**
**Argument:** Amal Clooney       **Role:**Spouse    **MannalAnotate[Y/N]**:
**Argument:** George Clooney   **Role:**Spouse    **MannalAnotate[Y/N]:**

##002  She was [**married**] to the cinematographer [**Theo Nischwitz**] and was sometimes credited as [**Gertrud Hinz-Nischwitz**].
**Trigger:** married    **Event Type:** Marriage       **MannalAnotate[Y/N]:**
**Argument:** Theo Nischwitz   **Role:**Spouse    **MannalAnotate[Y/N]:**
**Argument:** Gertrud Hinz-Nischwitz   **Role:**Spouse    **MannalAnotate[Y/N]:**

Figure 5: Examples of manual evaluations.

We firstly manually evaluate the precision of our automatically generated labeled data. We randomly select 500 samples from our automatically labeled data. Each selected sample is a sentence with a highlighted trigger, labeled arguments and corresponding event type and argument roles. Figure 5 gives some samples. Annotators are asked to assign one of two labels to each sample. "Y": the word highlighted in the given sentence indeed triggers an event of the corresponding type or the word indeed plays the corresponding role in that event. Otherwise "N" is labeled. It is very easy to annotate a sample for annotators, thus the annotated results are expected to be of high quality. Each sample is independently annotated by three annotators[6] (including one of the authors and two of our colleagues who are familiar with event extraction task) and the final decision is made by voting.

We repeat above evaluation process on the final 72, 611 labeled data three times and the average

---

[6]The inter-agreement rate is 87.5%

| Stage | Average Precision |
|---|---|
| Trigger Labeling | 88.9 |
| Argument Labeling | 85.4 |

Table 3: Manual Evaluation Results

precision is shown in Table 3. Our automatically generated data can achieve a precision of 88.9 and 85.4 for trigger labeling and argument labeling respectively, which demonstrates that our automatically labeled data is of high quality.

## 5.3 Automatic Evaluations of Labeled Data

To prove the effectiveness of the proposed approach automatically, we add automatically generated labeled data into ACE dataset to expand the training sets and see whether the performance of the event extractor trained on such expanded training sets is improved. In our automatically labeled data, there are some event types that can correspond to those in ACE dataset. For example, our *people.marriage* events can be mapped to *life.marry* events in ACE2005 dataset. We mapped these types of events manually and we add them into ACE training corpus in two ways. (1) we delete the human annotated ACE data for these mapped event types in ACE dataset and add our automatically labeled data to remainder ACE training data. We call this Expanded Data (ED) as *ED Only*. (2) We directly add our automatically labeled data of mapped event types to ACE training data and we call this training data as *ACE+ED*. Then we use such data to train the same event extraction model (DMCNN) and evaluate them on the ACE testing data set. Following (Nguyen et al., 2016; Chen et al., 2015; Li et al., 2013), we used the same test set with 40 newswire articles and the same development set with 30 documents and the rest 529 documents are used for ACE training set. And we use the same evaluation metric P, R, F as ACE task defined. We select three baselines trained with ACE data. (1) *Li's structure*, which is the best reported structured-based system (Li et al., 2013). (2) *Chen's DMCNN*, which is the best reported CNN-based system (Chen et al., 2015). (3) *Nguyen's JRNN*, which is the state-of-

| Methods | Trigger Identification(%) | | | Trigger Identification + Classification(%) | | | Argument Identification(%) | | | Argument Role(%) | | |
|---|---|---|---|---|---|---|---|---|---|---|---|---|
| | P | R | F | P | R | F | P | R | F | P | R | F |
| Li's structure trained with ACE | 76.9 | 65.0 | 70.4 | 73.7 | 62.3 | 67.5 | 69.8 | 47.9 | 56.8 | 64.7 | 44.4 | 52.7 |
| Chen's DMCNN trained with ACE | 80.4 | 67.7 | 73.5 | 75.6 | 63.6 | 69.1 | 68.8 | 51.9 | 59.1 | 62.2 | 46.9 | 53.5 |
| Nguyen's JRNN trained with ACE | 68.5 | 75.7 | 71.9 | 66.0 | 73.0 | 69.3 | 61.4 | 64.2 | 62.8 | 54.2 | 56.7 | 55.4 |
| DMCNN trained with ED Only | 77.6 | 67.7 | 72.3 | 72.9 | 63.7 | 68.0 | 64.9 | 51.7 | 57.6 | 58.7 | 46.7 | 52.0 |
| DMCNN trained with ACE+ED | 79.7 | 69.6 | **74.3** | 75.7 | 66.0 | **70.5** | 71.4 | 56.9 | **63.3** | 62.8 | 50.1 | **55.7** |

Table 4: Overall performance on ACE blind test data

the-arts system (Nguyen et al., 2016).

The results are shown in Table 4. Compared with all models, *DMCNN trained with ACE+ED* achieves the highest performance. This demonstrates that our automatically generated labeled data could expand human annotated training data effectively. Moreover, compared with *Chen's DMCNN trained with ACE*, *DMCNN trained with ED Only* achieves a competitive performance. This demonstrates that our large scale automatically labeled data is competitive with elaborately human-annotated data.

### 5.4 Discussion

**Impact of Key Rate**

In this section, we prove the effectiveness of KR to find key arguments and explore the impact of different numbers of key arguments to automatically generate data. We specifically select two methods as baselines for comparison with our KR method: ER and RS, which use the event relevance and role salience to sort arguments of each type of events respectively. Then we choose the same number of key arguments in all methods and use these key arguments to label data. After that we evaluate these methods by using above automatic evaluations based on ACE data. Results are shown in Table 5. *ACE+KR* achieve the best performance in both stages. This demonstrates the effectiveness of our KR methods.

| Feature | Trigger $F_1$ | Argument $F_1$ |
|---|---|---|
| ACE | 69.1 | 53.5 |
| ACE + RS | 70.1 | 55.3 |
| ACE + ER | 69.5 | 54.2 |
| ACE + KR | **70.5** | **55.7** |

Table 5: Effects of ER, RS and KR

To explore the impact of different numbers of key arguments, we sort all arguments of each type of events according to KR value and select top $k$ arguments as the key arguments. Examples are shown in Table 2. Then we automatically evaluate the performance by using automatic evaluations proposed above. Figure 6 shows the results, when we set $k = 2$, the method achieves a best

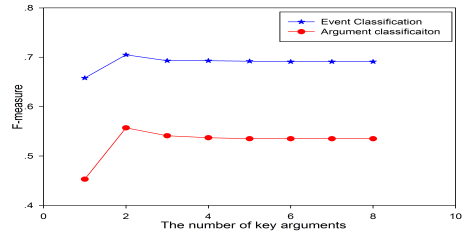

Figure 6: Effects of the number of key arguments

performance in both stages. Then, the F1 value reduces as $k$ grows. The reason is that the heuristics for data labeling are stricter as $k$ grows. As a result, less training data is generated. For example, if $k = 2$, we will get $25,797$ sentences labeled as *people.marriage* events and we will get $534$ labeled sentences, if $k = 3$. However, when we set $k = 1$, although more labeled data are generated, the precision could not be guaranteed.

**Impact of Trigger Rate and FrameNet**

In this section, we prove the effectiveness of TR and FrameNet to find triggers. We specifically select two methods as baselines: TCF and TETF. TCF, TETF and TR respectively use the trigger candidate frequency, trigger event type frequency and trigger rate to sort trigger candidates of each type of events. Then we generate initial trigger lexicon by using all trigger candidates with high TCF value, TETF value or TR value. We set these hyper parameters as $0.8, 0.9$ and $0.8$, respectively, which are determined by grid search from $(0.5, 0.6, 0.7, 0.8, 0.9, 1.0)$. FrameNet was used to filter noisy verbal triggers and expand nominal triggers. Trigger examples generated by *TR+Framenet* are shown in Table 2. Then we evaluate the performance of these methods by using above automatic evaluations. Results are shown in Table 6, Compared with *ACE+TCF* and *ACE+TETF*, *ACE+TR* gains a higher performance in both stages. It demonstrates the effectiveness of our TR methods. When we use FrameNet to generate triggers, compared with *ACE+TR*, we get a $1.0$ improvement on trigger classification and a $1.7$ improvement on argument classification. Such improvements are higher than improvements gained by other methods (TCF, IEF, TR), which demon-

strates the effectiveness of the usage of FrameNet.

| Feature | Trigger | Argument |
|---|---|---|
| | $F_1$ | $F_1$ |
| ACE | 69.1 | 53.5 |
| ACE + TCF | 69.3 | 53.8 |
| ACE + TETF | 69.2 | 53.7 |
| ACE + TR | 69.5 | 54.0 |
| ACE + TR + FrameNet | **70.5** | **55.7** |

Table 6: Effects of TCF, TETF,TR and FrameNet

### 5.5 Performance of DMCNN-MIL

Following previous work (Mintz et al., 2009) in distant supervised RE, we evaluate our method in two ways: held-out and manual evaluation.

**Held-out Evaluation**
In the held-out evaluation, we hold out part of the Freebase event data during training, and compare newly discovered event instances against this held-out data. We use the following criteria to judge the correctness of each predicted event automatically: (1) An event is correct if its key arguments and event type match those of an event instance in Freebase; (2) An argument is correctly classified if its event type and argument role match those of any of the argument instance in the corresponding Freebase event. Figure 7 and Figure 8 show the precision-recall (P-R) curves for each method in the two stages of event extraction respectively. We can see that multi-instance learning is effective to alleviate the noise problem in our distant supervised event extraction.

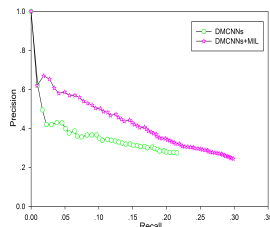 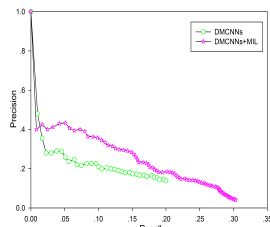

Figure 7: P-R curves for event classification.　Figure 8: P-R curves for argument classification.

**Human Evaluation**
Because the incomplete nature of Freebase, held-out evaluation suffers from false negatives problem. We also perform a manual evaluation to eliminate these problems. In the manual evaluation, we manually check the newly discovered event instances that are not in Freebase. Because the number of these event instances in the test data is unknown, we cannot calculate the recall in this case. Instead, we calculate the precision of the top n extracted event instances. The human evaluation results are presented in Table 7. We can see that DMCNNs-MIL achieves the best performance.

| Methods | Event Classificaiton | | | |
|---|---|---|---|---|
| | Top 100 | Top 300 | Top 500 | Average |
| DMCNNs | 58.7 | 54.3 | 52.9 | 55.3 |
| **DMCNNs+MIL** | **70.6** | **67.2** | **64.3** | **67.4** |
| Methods | Argument Classificaiton | | | |
| | Top 100 | Top 300 | Top 500 | Average |
| DMCNNs | 43.5 | 40.6 | 36.7 | 40.3 |
| **DMCNNs+MIL** | **50.8** | **45.6** | **43.5** | **46.6** |

Table 7: Precision for top 100, 300, and 500 events

## 6 Related Work

Most of previous event extraction work focused on supervised learning paradigm and trained event extractors on human-annotated data which yield relatively high performance. (Ahn, 2006; Ji and Grishman, 2008; Hong et al., 2011; McClosky et al., 2011; Li et al., 2013, 2014; Chen et al., 2015; Nguyen and Grishman, 2015; Nguyen et al., 2016). However, these supervised methods depend on the quality of the training data and labeled training data is expensive to produce. Unsupervised methods can extract large numbers of events without using labeled data (Chambers and Jurafsky, 2011; Cheung et al., 2013; Huang et al., 2016). But extracted events may not be easy to be mapped to events for a particular knowledge base.

Distant supervision have been used in relation extraction for automatically labeling training data (Mintz et al., 2009; Hinton et al., 2012; Krause et al., 2012; Krishnamurthy and Mitchell, 2012; Berant et al., 2013; Surdeanu et al., 2012; Zeng et al., 2015). But DS for RE cannot directly use for EE. For the reasons that an event is more complicated than a relation and the task of EE is more difficult than RE. The best reported supervised RE and EE system got a F1-score of $88.0\%$ (Wang et al., 2016) and $55.4\%$ (Nguyen et al., 2016) respectively. Reschke et al. (2014) extended the distant supervision approach to fill slots in plane crash. However, the method can only extract arguments of one plane crash type and need flight number strings as input. In other words, the approach cannot extract whole event with different types automatically.

## 7 Conclusion and Future Work

In this paper, we present an approach to automatically label training data for EE. The experimental results show the quality of our large scale automatically labeled data is competitive with elaborately human-annotated data. Also, we provide a DMCNN-MIL model for this data as a baseline for further research. In the future, we will use the proposed automatically data labeling method to more event types and explore more models to extract events by using automatically labeled data.

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
