# Peer review of "Automatically Labeled Data Generation for Large Scale Event Extraction"

_ACL 2017 — decision unknown_

[Official Review · Reviewer 1 · rating 3 · confidence 4]
soundness 5 · originality 5 · clarity 4 · impact 3 · substance 4 · appropriateness 5 · meaningful comparison 3 · presentation format Poster

- Strengths:

Improves over the state-of-the-art. Method might be applicable for other
domains.

- Weaknesses:

Not much novelty in method.  Not quite clear if data set is general enough for
other domains.

- General Discussion:

This paper describes a rule-based method for generating additional
weakly labeled data for event extraction.  The method has three main
stages.  First, it uses Freebase to find important slot fillers
for matching sentences in Wikipedia (using all slot fillers is too
stringent resulting in too few matches).  Next, it uses FrameNet to
to improve reliability of labeling trigger verbs and to find nominal
triggers.  Lastly, it uses a multi-instance learning to deal with
the noisily generated training data.

What I like about this paper is that it improves over the
state-of-the-art on a non-trival benchmark.  The rules involved
don't seem too obfuscated, so I think it might be useful for the
practitioner who is interested to improve IE systems for other domains.  On
the other hand, some some manual effort is still needed, for example for
mapping Freebase
event types to ACE event types (as written in Section 5.3 line 578).  This also
makes it difficult for future work to calibrate apple-to-apple against this
paper.              Apart
from this, the method also doesn't seem too novel.

Other comments:

- I'm also concern with the generalizability of this method to other
  domains.  Section 2 line 262 says that 21 event types are selected
  from Freebase.  How are they selected?  What is the coverage on the 33 event
types
in the ACE data.

- The paper is generally well-written although I have some
  suggestions for improvement.              Section 3.1 line 316 uses "arguments
liked time, location...".  If you mean roles or arguments, or maybe
you want to use actual realizations of time and location as
examples.  There are minor typos, for e.g. line 357 is missing a
"that", but this is not a major concern I have for this paper.